# The Diagnostic Value of ACSL1, ACSL4, and ACSL5 and the Clinical Potential of an ACSL Inhibitor in Non-Small-Cell Lung Cancer

**DOI:** 10.3390/cancers16061170

**Published:** 2024-03-16

**Authors:** Yunxia Ma, Miljana Nenkov, Alexander Berndt, Mohamed Abubrig, Martin Schmidt, Tim Sandhaus, Otmar Huber, Joachim H. Clement, Susanne M. Lang, Yuan Chen, Nikolaus Gaßler

**Affiliations:** 1Section Pathology of the Institute of Forensic Medicine, Jena University Hospital, Friedrich Schiller University Jena, Am Klinikum 1, 07747 Jena, Germany; 2Institute of Biochemistry II, Jena University Hospital, Friedrich Schiller University Jena, Nonnenplan 2, 07747 Jena, Germany; 3Clinic of Cardiothoracic Surgery, Jena University Hospital, Friedrich Schiller University Jena, Am Klinikum 1, 07747 Jena, Germany; 4Department of Hematology and Medical Oncology, Jena University Hospital, Am Klinikum 1, 07747 Jena, Germany; 5Department of Internal Medicine V, Jena University Hospital, Friedrich Schiller University Jena, Am Klinikum 1, 07747 Jena, Germany; susanne.lang@med.uni-jena.de

**Keywords:** ACSLs, ACSL inhibitor, lung cancer, diagnostic marker, combination therapy

## Abstract

**Simple Summary:**

Long-chain Acyl-CoA synthetase (ACSL) family members are involved in long-chain fatty acid activation, a crucial step for cells to utilize fatty acids (FAs) in the regulation of FA homeostasis. Cancer cells utilize the limited environmental FAs via the dysregulation of ACSL isoforms to adapt to an altered lipid metabolism. Dysregulated ACSLs are associated with either cancer promotion or inhibition, dependent on the cellular context. The role of ACSLs in different cancer types has not yet been fully elucidated, and, particularly, their role in lung cancer is unclear. In this study, we analyzed the expression pattern of ACSLs, evaluated their diagnostic values, and explored the therapeutic potential of the ACSL inhibitor Triacsin C in human lung cancer cells.

**Abstract:**

Abnormal expression of ACSL members 1, 3, 4, 5, and 6 is frequently seen in human cancer; however, their clinical relevance is unclear. In this study, we analyzed the expression of ACSLs and investigated the effects of the ACSL inhibitor Triacsin C (TC) in lung cancer. We found that, compared to normal human bronchial epithelial (NHBE) cells, ACSL1, ACSL4, and ACSL6 were highly expressed, while ACSL3 and ACSL5 were lost in the majority of lung cancer cell lines. ACSL activity was associated with the expression levels of the ACSLs. In primary lung tumors, a higher expression of ACSL1, ACSL4, and ACSL5 was significantly correlated with adenocarcinoma (ADC). Moreover, ACSL5 was significantly reversely related to the proliferation marker Ki67 in low-grade tumors, while ACSL3 was positively associated with Ki67 in high-grade tumors. Combination therapy with TC and Gemcitabine enhanced the growth-inhibitory effect in EGFR wild-type cells, while TC combined with EGFR-TKIs sensitized the EGFR-mutant cells to EGFR-TKI treatment. Taken together, the data suggest that ACSL1 may be a biomarker for lung ADC, and ACSL1, ACSL4, and ACSL5 may be involved in lung cancer differentiation, and TC, in combination with chemotherapy or EGFR-TKIs, may help patients overcome drug resistance.

## 1. Introduction

Lung cancer remains continuously the foremost cause of cancer-related death worldwide [1,2]. Despite the great progress which has been made in lung cancer treatment, particularly in receptor tyrosine kinase inhibition-based targeted therapy and immune checkpoint blockade-based cancer immunotherapy, lung cancer survival has remained poor over the past two decades. A majority of patients who initially responded to therapy acquired drug resistance after treatment for 6 months to 1 year. Drug resistance remains a major clinical challenge in lung cancer. Metabolic adaptation is one of the important factors related to the maintenance of therapy resistance in cancer [3]. In addition to cancer cells excessively utilizing glucose (well-known as the “Warburg effect”) and glutamine to enhance their metabolic activity [4], they also use fatty acids (FAs) as a rich energy source. An abnormal FA metabolism, including enhanced de novo FA synthesis and FA oxidation, has been reported in different types of cancer, including lung cancer [5].

The conversion of FAs to activated acyl-CoAs catalyzed by acyl-CoA synthetases (ACSs) is the first and crucial step in the regulation of fatty acid homeostasis, which is involved in the structural synthesis of lipid derivatives, energy storage and production, as well as signaling transmission [6]. Long-chain Acyl-CoA synthetase (ACSL) family members are responsible for the activation of long-chain fatty acids (LCFAs) with aliphatic tails of 13 to 21 carbons. Lipid metabolic reprogramming through the regulation of ACSL metabolites is one of the remarkable features of cancer cells [7,8,9].

The ACSL family comprises five isoenzymes: ACSL1, ACSL3, ACSL4, ACSL5, and ACSL6. Each isoenzyme has tissue-specific expression patterns [10]. Dysregulated ACSL isoforms are common phenomena observed in multiple cancer types, according to the RNA-Seq data from the TNM Plot database [11,12]. Moreover, each ACSL family member has different effects on lung cancer. According to the results from the Oncomine and PrognoScan databases, ACSL1 is modestly expressed in lung carcinoma, negatively correlated with fatty acid transport protein 2 (FATP2), and the interaction between ACSL1 and FATP2 promotes non-small-cell lung cancer (NSCLC) cell progression [13]. ACSL3-mediated fatty acid oxidation is required for mutant KRAS lung tumorigenesis [8]. ACSL4 upregulated by bone morphogenetic protein 4 (BMP4) enhances the cancer cell fatty acid metabolism, which is associated with acquired drug resistance in EGFR-mutant NSCLC cells [9]. ACSL5 is highly expressed in lung adenocarcinoma (ADC), correlating with a better prognosis [14]. An in vitro study by Zhang et al. demonstrates that the upregulation of ACSL5 by lysophosphatidylcholine (LysoPC) suppresses lung cancer cell proliferation [15]. Although RNA-seq data from public databases are available regarding the expression of ACSLs in human cancer, protein expression of ACSLs in human lung cancer is still not well studied. Therefore, a systematic expression analysis of ACSLs and an evaluation of their clinical relevance are needed, which may help to achieve a better understanding of the biological function of ACSLs in lung cancer.

ACSLs modulate the lipid metabolism in cancer cells. Targeting the FA metabolism has been proposed as a promising therapeutic strategy [16]. Several ACSLs are highly expressed in cancer cells; thus, targeting ACSLs might implicate clinical application in the treatment of cancer. Triacsin C (TC), a fungal metabolite, is able to inhibit ACSL1, ACSL3, and ACSL4 directly [17]. TC also inhibits ACSL5 and ACSL6 at high concentrations over 10 µM and 50 µM, respectively [18]. The treatment of macrophages with TC and very-low-density lipoproteins leads to decreased triacylglyceride (TCG) accumulation but increased free fatty acid intracellular levels, which cause lipotoxicity characterized by apoptosis [19]. TC modulates the lipid metabolism by inhibiting lipid droplet formation and promoting mitochondrial biogenesis in rat liver [20]. Moreover, a high dose of TC suppresses tumor growth in a subset of p53-defective tumor cells [21]. Despite the fact that TC significantly blocks lipid droplet formation through the inhibition of ACSL isoforms, TC has a high cytotoxicity, which impedes drug development and its clinical use [22]. Combined treatment regimens are promising approaches to reduce side effects and increase drug efficacy, compared to monotherapy. TC at a non-toxic dose in combination with the chemotherapeutic drug etoposide induces cell death in a malignant glioma nude mouse xenograft model [23]. The application of tyrosine kinase inhibitors (TKIs) has largely improved the clinical outcome of patients with NSCLC with tumors harboring epidermal growth factor receptor (EGFR)-activating mutations. However, drug resistance is a big obstacle to their long-term therapy efficacy. Recently, it has been reported that the fatty acid synthase inhibitor decreases resistance features in EGFR-TKI-resistant NSCLC cell models [24]. The effectiveness of the combination therapy with TC and EGFR-TKI has not yet been explored in NSCLC cells.

In this study, we analyzed the expression of ACSL isoforms and ACSL activity in a panel of human lung cancer cell lines and assessed the clinical relevance of ACSLs in primary lung samples of patients with cancer. Moreover, the growth-inhibitory effects of TC combined with Gemcitabine or EGFR-TKIs were investigated in EGFR-wild-type or EGFR-mutant NSCLC cells.

## 2. Materials and Methods

### 2.1. Cell Lines and Cell Culture

Normal human bronchial epithelial (NHBE) cells were purchased from Lonza (Cologne, Germany) and grown in Bronchial Epithelial Growth media (Lonza, Cologne, Germany). A panel of lung cancer cell lines including one small-cell lung cancer (SCLC) cell line (COLO677), seven lung ADC cell lines (H1299, H2030, H23, A549, H322, H1650, and H1975), and three squamous cell carcinoma (SCC) cell lines (H2170, H226, and H157) were obtained from the American Type Culture Collection (ATCC, Rockville, MD, USA) and the German Collection of Microorganisms and Cell Culture (DSMZ, Braunschweig, Germany). The cells were cultured in RPMI 1640 medium (PAN-Biotech, Aidenbach, Germany) supplemented with 10% fetal bovine serum (Biochrom AG, Berlin, Germany), and maintained at 37 °C with 5% CO_2_.

### 2.2. RNA Extraction, Reverse Transcription (RT), and Quantitative PCR (qPCR)

RNA was isolated from the cells using the Trizol reagent (Qiagen, Hilden, Germany) according to the manufacturer’s instructions. Five hundred nanograms (ng) of RNA was used for cDNA synthesis using a QuantiTect Reverse Transcription Kit (Qiagen). Quantitative PCR was carried out using a FastStart Universal SYBR Green Master Mix (Roche, Mannheim, Germany). Glyceraldehyde 3-phosphate dehydrogenase (GAPDH) was applied as an internal control. Primer sequences of ACSL family members are shown in Appendix A.

### 2.3. Protein Isolation and Western Blotting (WB)

Proteins from the cell lysates were isolated using radioimmunoprecipitation assay (RIPA) buffer (Merck, Taufkirchen, Germany) supplemented with a cocktail of protease inhibitors (Merck, Taufkirchen, Germany) and PhosSTOP (Merck, Taufkirchen, Germery). The protein concentration was determined using a BCA kit (Thermo Scientific, Carlsbad, CA, USA) according to the manufacturer’s protocol. Protein samples (15 µg) with a loading buffer were loaded onto the SDS-polyacrylamide gel after denaturation at 95 °C for 5 min. After the electrophoresis and transfer of the proteins on 0.2 µm nitrocellulose membranes (GE Healthcare, Munich, Germany), the membranes were then blocked for 1 h with 5% (*w*/*v*) milk in 1xTris-buffered saline with 0.1% (*v*/*v*) Tween20 detergent (TBST) and incubated with primary antibodies overnight. After washing them three times with 1xTBST, the membranes were incubated with secondary antibodies for 1 h and washed again. Enhanced chemiluminescence (Santa Cruz Biotechnology, Heidelberg, Germany) was used for signal detection using G:BOX Chemi XX6 (Syngene, London, UK). Detailed information on the primary and secondary antibodies is listed in Appendix A.

### 2.4. Measurement of Total ACSLs Activity Using the Substrate [^3^H]-Palmitic Acid (PA)

The acyl-CoA synthetase (ACS) activity was determined by measuring the production of [^3^H]-palmitoyl-CoA using the protocol of Füllekrug and Poppelreuther with slight modifications [25]. Instead of using [^14^C]-oleic acid, we applied [9,10-^3^H(N)]-palmitic acid (PerkinElmer, Rodgau, Germany) as the substrate. Briefly, the samples were incubated in 100 mM Tris HCl (pH 7.4), 5 mM MgCl_2_, 200 μM DTT, 10 mM ATP, 200 μM CoA, 0.1% (*v*/*v*) Triton X-100, 19.8 μM palmitic acid, and 0.2 μM [9,10-^3^H(N)]-palmitic acid (0.5 Ci/mol) for 20 min at 37 °C. The reactions were stopped by adding 600 μL of Dole’s solution (Isopropanol: Heptane: H_2_SO_4_ 40:10:1). The unreacted palmitate was removed by four sequential heptane washes, and the final radioactivity in the aqueous phase containing palmitoyl-CoA was measured by liquid scintillation counting (Tri-Carb 2810 TR, PerkinELmer, Rodgau, Germany). The specific acyl-CoA synthetase activity was calculated as pmol palmityl-CoA/min/µg protein, considering the incubation time and the total protein content in the assay.

### 2.5. Multiplex Immunofluorescence (mIF)

The tissue distribution of ACSL family members in normal lung tissue was analyzed by mIF. Adjacent normal lung tissue more than 5 cm away from tumor lesions was prepared into a paraffin block. The ethical approval for the specimens in this study was been issued by the local ethical committee of University Hospital Jena (Reg.-Nr.: 2020-1987-Material). Sections (3 μm) were cut from the paraffin block, placed on glass slides, and heated at 60 °C for 3 h. The residual paraffin was dewaxed and gradually rehydrated. Antigen retrieval (AR) was performed using AR6 or AR9 according to the primary antibody at 96 °C for 25 min and cooled down to room temperature. Then, the slides were incubated with an antibody diluent/blocking solution. After removing the blocking solution, the slides were incubated with the primary antibody for 1 h. After removal of the unbound primary antibody by washing with 1xTBST, the slides were incubated with the secondary antibody (Opal polymer HPR Ms + Rb) and Opal fluorophore. Once the second round of antigen retrieval had been completed, the automatically repeated staining cycle was carried out, and the following antibodies in a specific order were given in different panels (Appendix A). Fluorescent slides were scanned by the Vectra^®^ Polaris™ Automated Quantitative Pathology Imaging System (AKOYA Biosciences, Marlborough, MA, USA) and viewed using Phenochart 1.1.0.

### 2.6. Immunohistochemistry (IHC)

A tissue microarray (TMA) containing 97 samples of primary lung tumor tissues was constructed. Two core biopsies (0.6 mm in diameter) were extracted from each sample. All the patients were surgically operated in the Hospital Bad Berka Germany from 1999 to 2002. Neither adjuvant radiotherapy nor chemotherapy was conducted before surgery. Ethical approval for the study was issued by the local ethical committee of University Hospital Jena (Nr: 3815-07/13). The preparation of the slides was similar to that for mIF. After antigen retrieval (sodium citrate, pH 6.0), the slide was blocked using the Biotin Blocking System (Agilent Technologies, Santa Clara, CA, USA) and then incubated with primary antibodies of ACSLs and Ki67 overnight (Appendix A). Detection was performed based on the protocol of the Dako REAL Detection System (Agilent Technologies). The IHC was scored semi-quantitatively as a score of 0 (<10% positively stained cells), a score of 1 (10–25% positively stained cells), a score of 2 (25–50% positively stained cells), and a score of 3 (>50% positively stained cells). For the statistical analysis, a score of 0 or 0–1 was considered a negative expression, while scores 1–3 or 2–3 were considered positive expressions.

### 2.7. Drug Treatment and Transfection

The stock solutions of Gemcitabine (8.3 nM, LC Laboratories, Hamburg, Germany) and TC (4.8 nM; BML-EI218-1000, Enzo Life Sciences, Lörrach, Germany) were prepared in 1xPBS and DMSO, respectively. The solutions (10 mM in DMSO for each) of EGFR-TKIs including Gefitinib, Afatinib, and Osimertinib were purchased from Selleckchem (Munich, Germany).

Stable transfection with the ACSL5 expression vector (CloneID:OHu02964) (GenScript, Piscataway, NJ, USA) and a control vector (pcDNA3.1^+^/C-(K)-DYK) was carried out in H1299 cells using the transfection reagent Lipofectamine^TM^ 2000 (Invitrogen, Carlsbad, CA, USA). Positive colonies were picked out based on the results of WB after 2 weeks of selection with 400 µg/µL Geneticin (G418) Sulfate (sc-29065) (Santa Cruz Biotechnology, Dallas, TX, USA). Stable transfection with ACSL5 shRNA (sc-60621-SH) and a control vector (sc-108060) was carried out in H1650 and H1975 cells, which showed endogenous expression of ACSL5. After incubation with 50 µg/mL of puromycin (sc-108071) (Santa Cruz Biotechnology, Dallas, TX, USA) for 2 weeks, the downregulation of ACSL5 was confirmed by RT-qPCR analysis.

Transient transfection with siRNAs (obtained from Santa Cruz Biotechnology, Dallas, TX, USA), including siRNA-ACSL1 (sc-60615), siRNA-ACSL4 (sc-60619), and control siRNA (sc-37007), was performed in H2170 cells. One day after transfection, part of the cells were harvested and reseeded in a 96-well plate for drug treatment, the rest of the cells remained in the culture for another 24 h and were collected for mRNA expression analysis.

### 2.8. Cell Viability Assay

The cells were seeded in 96-well plates (9 × 10^3^–1.5 × 10^4^ cells/well) and cultured until they reached 70% confluence. Serial concentrations of chemotherapeutic drug (Gemcitabine) or EGFR-TKIs (Gefitinib, Afatinib, and Osimertinib) were added and incubated for 48 h. PBS or DMSO was used as the control. After that, the culture medium was removed, and the cells were fixed with methanol for 10 min, stained with 0.1% (*w*/*v*) crystal violet for 10 min, and air-dried at room temperature after washing with 1xPBS. Before the measurement, the cells were incubated with 100 µL of 30% (*v*/*v*) acetic acid at room temperature for 1 h to solubilize the dye. The colorimetric absorbance was read at OD590 nm using a Multimode Microplate Reader (Infinite M200 Pro Tecan, Männedorf, Switzerland). The values were normalized to the DMSO- or PBS-treated control, and GraphPad Prism was used for plotting the graphs.

### 2.9. Data Analysis

Student’s *t*-test was used to analyze gene expression in the lung cancer cell lines compared to the NHBE ones after normalization with an internal control (GAPDH). A chi-square or Fisher’s exact test was performed to assess the association between the expression of ACSL family members and the clinicopathological parameters; the Kaplan–Meier survival curve was applied to evaluate the prognostic value of ACSLs; and the Pearson Chi-square test was applied to compare categorical variables, using the software program IBM SPSS Statistics 21. A two-way Anova or Student’s *t*-test was applied to analyze the cell viability, and the diagrams were created using GraphPad Prism 9. Statistical significance was defined as a two-tailed *p*-value less than 0.05.

## 3. Results

### 3.1. Expression and Enzymatic Activity of ACSL Isoforms in Normal Human Bronchial Epithelium (NHBE) Cells and Lung Cancer Cell Lines

To investigate the expression of ACSL isoforms in NHBE cells and a panel of lung cancer cell lines including three SCC, six ADC, and one SCLC, RT-qPCR and Western blot (WB) analyses were carried out. Compared to the NHBE cells, the mRNA expression of ACSL1, ACSL3, and ACSL6 was downregulated in all the lung cancer cell lines. ACSL4 and ACSL5 were downregulated in the majority of lung cancer cell lines, except for an upregulated ACSL4 in H226, H2030, and H1975 and increased ACSL5 levels in COLO677, H322, H1650, and H1975, compared to the NHBE cells (Figure 1A).

On the protein level, as shown in Figure 1B (for the original Western blot, see Appendix A), ACSL1, ACSL4, and ACSL6 were widely expressed in the lung cancer cell lines that we tested, and, compared to the NHBE cells, they were upregulated in the majority of the cancer cell lines. ACSL3 was expressed in COLO677, A549, H322, H1650, and H1975, while ACSL5 was expressed in H2170, COLO677, A549, H1650, and H1975. Additionally, various splicing variants of ACSL isoforms, except for ACSL4, were observed. ACSL1 had three variants, and the composition of ACSL1 variants between the NHBE and lung cancer cell lines was different. Two variants of ACSL3, ACSL5, and ACSL6 were observed.

The activity of ACSLs was measured using [^3^H]-palmitic acid as a substrate. Compared to the NHBE cells, ACSL activity was higher in four lung cancer cell lines, including A549, H322, H1650, and H1975, in which most of the ACSLs were expressed (Figure 1C).

Taken together, our data show that ACSL family members have different expression patterns and that somatic mutations, epigenetic mechanisms, and the post-transcriptional regulation of ACSLs might play a role in lung cancer cells. Additionally, the dysregulated expression of ACSLs was associated with an altered ACSL activity in lung cancer cells.

### 3.2. Tissue Distribution of ACSL Isoforms in Normal Human Lung Tissues

To examine the distribution of ACSL isoforms in normal lung tissue, mIF was performed. The pneumocyte type II marker surfactant protein C (SFPC) and the epithelium marker cytokeratin (CK) were included. As shown in Figure 2, SFPC defines the alveolar type II cells and CK marks the epithelium, including the small airway epithelium (bronchial epithelium) and distal lung epithelium (alveolar epithelium) in which pneumocyte type I/II cells play a key role. We observed that all ACSLs were expressed in normal bronchial and alveolar epithelia (Figure 2, Appendix A). Using the marker SFPC for the type II pneumocytes, we observed positive staining for ACSLs in the pneumocytes in which lipoprotein complexes, the surfactant proteins, are synthesized and secreted. The wide expression of ACSLs in the alveoli indicates that ACSL family members might be required to maintain the normal structure and function of the lungs. In addition, strong staining for ACSL1, ACSL3, and ACSL6 was observed in macrophages.

### 3.3. The Correlation between the Expression of ACSL Isoforms and the Clinicopathological Parameters in Primary Lung Tumor Tissues

To evaluate the association between the protein expression of ACSL isoforms and the clinicopathological parameters in primary lung tumor tissues, conventional IHC was carried out on a tissue microarray (TMA). The representative staining of ACSL isoforms is shown in Figure 3. As listed in Table 1, the higher expression of ACSL1, ACSL4, and ACSL5 was significantly correlated with lung ADC, compared to SCC (ACSL1, *p* = 0.011; ACSL4, *p* = 0.021; ACSL5, *p* = 0.000), and, particularly, a majority (43/47) of ADC samples exhibited positive staining for ACSL1. A lower ACSL3 expression was statistically linked to age (60 years or more, *p* = 0.042). In addition, a higher expression of ACSL1 and ACSL5 was significantly related to a low tumor grade (ACSL1, *p* = 0.023; ACSL5, *p* = 0.025), while a higher expression of ACSL4 was significantly associated with a high grade (*p* = 0.009).

Moreover, we analyzed the correlation between tumor grade and ACSL expression in a subtype of lung cancer. It turned out that a higher expression of ACSL1, ACSL3, and ACSL5 was significantly correlated with a lower-grade ADC (ACSL1, *p* = 0.026; ACSL3, *p* = 0.027; ACSL5, *p =* 0.03), while a higher expression of ACSL4 was significantly associated with a higher-grade ADC (*p =* 0.024) (Table 2).

Data from the public database Kaplan–Meier Plotter (KM Plotter) [26] shows that a high mRNA expression of ACSL1 and ACSL5 but not ACSL3, ACSL4, or ACSL6 is significantly associated with a longer overall survival (Appendix A). However, in our study, the survival analysis did not reveal a significant correlation between ACSLs’ protein expression and clinical outcomes.

Additionally, we analyzed the correlation among ACSL isoforms in the fifty samples in which all the ACSLs were expressed. We found that ACSL1 was significantly positively correlated with ACSL5 (r = 0.295, *p* = 0.044), while ACSL3 and ACSL5 were significantly negatively associated with ACSL6 (ACSL3 vs. ACSL6, r = −0.28, *p* = 0.048; ACSL5 vs. ACSL6, r = −0.314, *p* = 0.026) (Table 3).

Taken together, our data suggest that ACSL1 may be a potential diagnostic marker for lung ADC, and that ACSL1, ACSL4, and ACSL5 may be involved in lung cancer differentiation.

To further explore whether ACSLs may influence tumor cell growth, the tumor proliferation marker Ki67 was included for the IHC analysis (Appendix A). We found that ACSL5 was significantly inversely related to Ki67 in low-grade tumors (r = −0.36, *p* = 0.02), while ACSL3 was positively associated with Ki67 in high-grade tumors (r = 0.397, *p* = 0.047) (Table 4). The data suggest that ACSL5 may exert a suppressive function in low-grade lung tumors, while ACSL3 may enhance tumor proliferation in high-grade tumors.

### 3.4. ACSL Enzymatic Activity after ACSL Inhibitor (Triacsin C) Treatment

To explore the potential therapeutic role of the ACSL inhibitor Triacsin C (TC), we treated lung cancer cell lines with different concentrations of TC. TC showed a strong effect in H1650 cells with 2 µM, 4 µM, and 8 µM of TC inhibiting the total ACSL activities to 55%, 53%, and 34%, respectively. However, in H1975 cells treated with the same concentrations of TC, we found that 83%, 78%, and 71% of the total ACSL activity still remained (Figure 4A, upper panel). In five lung cancer cell lines, including H2170, H1299, H226, A549, and H157, we found that TC treatment (8 µM) led to a significantly decreased ACSL activity, compared to DMSO treatment (Figure 4A, lower panel). In H1299 cells transfected with ACSL5, one of the ACSL family members, TC treatment resulted in significantly reduced ACSL activity, compared to DMSO treatment; however, compared to the mock cells, ACSL activity was still higher in the ACSL5 transfected cells (Figure 4B, upper panel). Consistent with this, the stable knockdown of ACSL5 led to a significantly decreased ACSL activity, with 25% of ACSL activity being eliminated compared to the shRNA control (Figure 4B, lower panel). The overexpression and knockdown of ACSL5 were validated using WB and RT-qPCR (Appendix A).

Additionally, we analyzed the effect of TC on cell viability in lung cancer cell lines. The dose–response curves indicated that 8 µM of TC was sufficient to reduce cell viability to 50% in H322 and A549, whereas a higher dose of TC to reach a 50% reduction in viability was observed in three other cell lines, 16 µM for H2170, 24 µM for H1975 and over 24 µM for H157. Moreover, an inhibition of 40% in terms of cell growth was observed with 24 µM of TC in H157 (Figure 4C, upper and lower panels). In all the cells, the data indicate that a low dose of TC (8 µM) could successfully suppress ACSL activity in the five lung cancer cell lines, but its effect on growth was varied, achieving a 50% reduction in growth only in H322 and A549.

### 3.5. The Effect of Triacsin C Combined with Chemotherapy/Targeted Therapeutic Drugs

It has been reported that a low dose of TC had no effect on tumor growth in a nude mouse xenograft model, but, when combined with a low dose of etoposide, it exerted an antitumor activity [23]. To explore the efficacy of TC in combination with chemotherapeutic drugs, a cell viability assay was carried out. Compared to Gemcitabine alone, combination therapy with TC (2 µM or 4 µM) and Gemcitabine significantly augmented the efficacy and increased the potency of TC in H157, A549, and H2170 cells (Figure 5A).

H1975 harbors EGFR mutations (L858R, T790M), and H1650 contains an E746-A750 deletion mutation in exon 19 of EGFR, together with a deletion in PTEN, and these two cell lines are resistant to first-generation EGFR-TKIs such as Gefitinib and Erotinib [27,28]. Therefore, we tested combination therapy with TC and EGFR-TKIs such as Gefitinib, Afatinib, and Osimertinib in lung cancer cell lines. Gefitinib combined with 8 µM or 4 µM of TC significantly reduced cell viability to 50% in the presence of Gefitinib at 13 µM or 15 µM concentrations in H1650 (Figure 5B). In H1975, Afatinib, in combination with TC (8 µM or 4 µM), inhibited cell growth at lower concentrations of 280 nM or 390 nM, compared to Afatinib treatment alone. There was a slight growth inhibition upon the combination of Osimertinib plus TC, compared to the treatment with Osimertinib alone.

### 3.6. The Effects of ACSL1, ACSL4, and ACSL5 Knockdown on Cell Viability

TC has been known as an inhibitor of ACSL isoforms [19]. To determine whether the downregulation of ACSL isoform alone has an effect on cell viability, transient transfection of siRNA/ACSL1 or siRNA/ACSL4 was performed in H2170. The gene silencing of ACSL1 (Appendix A) or ACSL4 (Appendix A) was confirmed by RT-qPCR. We observed that a 50% reduction in cell viability was achieved in ACSL1- and ACSL4-knockdown cells treated with 10 nM Gemcitabine (Figure 6A). Moreover, the stable knockdown of ACSL5 was established in H1650 and confirmed by RT-qPCR (Appendix A, and for the original Western blot, see Appendix A). The knockdown of ACSL5 reduced cell viability to 50% in the presence of 35 nM of Gemcitabine (Figure 6B, left) as well as 14 µM of Gefitinib, respectively (Figure 6B, right).

## 4. Discussion

ACSL isoforms are involved in long-chain FA activation, a rate-limiting step for organisms to utilize free long-chain FAs from exogenous or endogenous sources to meet diverse body needs [6]. It has been depicted that each ACSL isoform has distinct preferred substrates [29]. The dysregulation of ACSLs can be frequently found in cancer cells, in line with the fact that cancer cells utilize the limited environmental FAs to adapt to the FA metabolism [9,30]. The role of ACSLs in lung cancer development and progression has not yet been clarified.

ACSL isoforms have tissue-specific expression patterns according to the RNA-Seq data of The Human Protein Atlas from 55 types of tissues/organs [31] (Appendix A). In normal lung tissue, we found that ACSLs showed a similar distribution/expression pattern. Particularly, the positive expression of ACSLs, mainly ACSL1, ACSL3, and ACSL5, was observed in alveolar type II epithelial cells. These type II cells are the most active metabolic cells in the alveoli, synthesizing surfactant lipids including dipalmitoylphosphatidylcholine, the main constituent of pulmonary surfactants [32]. Given the fact that ACSLs are highly expressed in alveolar type II cells and that they play an important role in the lipid metabolism, we speculate that ACSLs may be required for maintaining alveolar function.

The dysregulated expression of ACSL isoforms has been found in multiple types of cancer according to the data from the TNM Plot [11] (Appendix A). In human lung cancer cells, we found that the proteins of ACSL1, ACSL4, and ACSL6 were highly expressed in most of the lung cancer cell lines observed, while the proteins of ACSL3 and ACSL5 were lost in the majority of the lung cancer cell lines, compared to the NHBE cells. The distinct mRNA and protein levels of ACSLs might be associated with somatic mutations, epigenetic mechanisms, and post-transcriptional regulation, which are frequently found in many biological systems [33,34,35]. The lung cancer cell lines, including A549, H322, H1650, and H1975, with endogenous expression of most of the ACSL family members exhibited a higher ACSL activity. However, the SCLC cell line COLO677, expressing all the ACSL isoforms, showed a lower ACSL activity compared to the NHBE cells. This indicates that, except for the expression levels, other factors might also influence ACSL activity. For example, the genetic background of cancer cells might influence enzymatic activity. It was found that the composition and enzymatic activities of the lactate dehydrogenase complex were altered in p53-mutant mice to favor pyruvate generation and hinder lactate production [36]. In our case, only palmitic acid was applied as a substrate for the measurement. It might be interesting in future work to clarify the ACSL activities using other important substrates including arachidonic acid (20:4), oleic acid (18:1), linoleic acid (18:2), and stearic acid (18:0) to better understand the effects of altered ACSL expression on tumor cell metabolism.

Alternative splicing produces ACSL transcript variants [37]. Splice variants may serve as cancer biomarkers [38]. ACSL1 has various transcripts among different organisms [39], and ACSL1 variants exhibit distinct functions in sheep lipid metabolism [40]. The results from the Western blot analysis indicate that normal bronchial epithelial cells and lung cancer cells may have different splice variants of ACSL1; however, it is not yet clear whether or to which extent this discrepancy may cause any biological impact on tumor cells. In primary lung tumors, a higher expression of ACSL1 was significantly associated with a low tumor grade in lung ADC, suggesting that ACSL1 is involved in the differentiation of lung ADC. An increased expression of ACSL1 has been found in different cancer types, associated with oncogenic functions in liver, colorectal, prostate, breast, and ovarian cancers [41,42,43,44,45]. Additionally, ACSL1 also plays an important role in cardiac and inflammatory diseases [46,47].

ACSL5 has mainly three protein variants, including a long form (739 aa), a short form (683 aa), and the exon 20-skipping form (659 aa), which are associated with two single-nucleotide polymorphisms (SNPs) [39,48,49]. In our study, we observed two protein variants (short and long) of ACSL5 in H1650 and H1975, the two EGFR-mutant cell lines. The upregulation of ACSL5 in the two lung cancer cell lines harboring EGFR mutations indicates a potential link between ACSL5 overexpression and EGFR mutations. In primary tumors, a higher expression of ACSL5 was markedly correlated with low-grade ADC, and, moreover, ACSL5 was negatively significantly associated with the expression of the proliferation marker Ki67, implying that ACSL5 may inhibit cell proliferation and enhance differentiation in lung ADC. A body of evidence demonstrates that ACSL5 protein variants have distinct functions in the aspect of regulating enterocyte and hepatocellular apoptosis, cell viability, diet-induced weight loss, as well as lipid metabolism [7,50,51,52,53]. In human cancer, ACSL5 shows pro- or anti-carcinogenic activity. It exerts a tumor-suppressive function in colorectal cancer and liver cancer, while it acts as an oncogene in malignant glioma [52,54,55].

Two ACSL3 transcripts were observed in the NHBE cells and have also been detected in mammary carcinoma cells as well as hamster liver and rat pancreatic islets [56,57]. Only one protein isoform of ACSL3 is present in human liver and pancreatic islets [58]. Similar to the expression pattern of ACSL5, ACSL3 was lost in most of the lung cancer cell lines, and, based on the expression of ACSL3 in primary lung tumor, we consider that ACSL3 might be involved in lung ADC differentiation. Moreover, ACSL3 expression is associated with a worse clinical outcome in patients with high-grade NSCLC. The data suggest that ACSL3 may play different roles in lung tumors with different levels of malignancy.

Two splice variants were identified in ACSL4. ACSL4 variant 1 is a ubiquitous variant of ACSL4, and variant 2 seems to be restricted to the brain [59,60]. ACSL4 variants have similar affinities for various unsaturated FA, but the reaction rates of each unsaturated FA are different [61]. ACSL4 was universally expressed in the lung cancer cell lines in our study. The significant correlation between ACSL4 protein expression and a higher tumor grading in lung ADC indicates its role in ADC differentiation. ACSL4 was upregulated in EGFR-mutant lung ADC cell lines and the upregulation of ACSL4 led to a higher energy metabolism in EGFR-mutant cells, enabling cancer cells to enhance cell growth and acquire drug resistance [9,62]. A study by Zhang et al. revealed an opposing role of ACSL4 in lung ADC cells, in which ACSL4 inhibited tumor growth via the induction of ferroptosis [30].

Two variants of ACSL6 appeared in the lungs, and more ACSL6 isoforms have been detected in E.coli, the human erythroleukemic cell line K562, and cDNA prepared from the bone marrow of patients suffering from leukemia [39]. ACSL6 is highly enriched in the brain and expressed in normal lung tissue. The overexpression of ACSL6 increases docosahexaenoic acid (DHA) and arachidonic acid internalization during neuronal differentiation [63]. However, the role of ACSL6 in lung cancer is poorly understood.

ACSLs are prognostic markers in cancer. The mRNA data from KM Plotter showed that ACSL1 and ACSL5 were favorable prognostic markers in lung cancer [58]. Moreover, the data from The Human Protein Atlas indicate that ACSL1, ACSL4, and ACSL5 are favorable prognostic markers in renal cancer, urothelial cancer, and endometrial cancer. ACSL3 is an unfavorable prognostic marker in liver cancer and lung cancer. ACSL6 is a favorable prognostic marker in leukemia [13].

Targeting ACSLs could inhibit tumor growth and enhance the chemosensitivity of cancer cells in ACSL-dependent tumorigenesis models [8,23,64]. A high dose of TC (30 mg/kg) has been shown to significantly reduce the tumor volume in xenograft nude mice models by inducing apoptosis [21]. In this study, we observed that different lung cancer cell lines responded differently to TC treatment. To avoid the toxicity of TC at high doses, combination regimens were applied. In EGFR-WT cell lines, combination therapy increased the sensitivity of EGFR-WT cell lines to the chemotherapeutic drug Gemcitabine. In EGFR-TKI-resistant cell lines, EGFR-TKIs, in combination with TC, enhanced the therapy’s efficacy. Therefore, a TC combination regimen with chemotherapeutic drugs or EGFR-targeted therapeutics might help patients overcome drug resistance in lung cancer cells. In addition, the specific knockdown of ACSL1, ACSL4, and ACSL5 also enhanced drug sensitivity to Gemcitabine and Gefitinib.

Cancer cells, by utilizing the surrounding glucose, glutamine, and lipids, facilitate their proliferation and survival [65]. An enhanced lipid metabolism accompanied by upregulated ACSLs has been seen in many cancer types [6].Growing evidence indicates that the pharmacological inhibition of ACSLs regulates lipid metabolism and includes apoptosis and ferroptosis in different cancer types. Rosiglitazone, an antagonist of peroxisome proliferator-activated receptors (PPARs), affects lipid composition through the transcriptional regulation of ACSL3 [66], and TC analogues inhibit rotavirus replication by regulating ACSL activity and lipid metabolism [67]. It has been shown that TC could enhance the etoposide-induced intrinsic apoptosis that is suppressed by the overexpression of ACSL5 in glioma cells [23]. Elevated ACSL4 levels, accompanied by increased mitochondrial phospholipid biosynthesis and fatty acid oxidation, promote anti-apoptosis in chemoresistant breast cancer, and ACSL4 silencing suppresses tumor growth and induces apoptosis [68]. Moreover, ACSL1, ACSL3, and ACSL4 are ferroptosis inducers. The inhibition of ACSL1-related ferroptosis by TC could restrain the hyper-inflammation induced by coronavirus infection [69]. The decrease in ACSL3 activity by TC suppresses erastin-induced ferroptosis [70]. The selective inhibition of ACSL4 by thiazolidinediones (TZDs), other PPAR agonists, prevents RSL3 (ferroptocide)-induced ferroptosis and lipid peroxidation [71]. Since ACSL4 promotes ferroptosis in lung cancer cells, it can be speculated that the inhibition of ACSL4 by TC might result in reduced levels of ferroptosis [30]. Therefore, clarifying the role of ACSL isoforms in cancer might help researchers in designing ACSL-targeted drugs and effective combination regimens for the treatment of lung cancer.

## 5. Conclusions

ACSL1, ACSL3, and ACSL5 are expressed in pneumocytes of the lung alveolar epithelium, suggesting their role in maintaining lung structure and function. In primary lung tumors, ACSL1 is a potential diagnostic marker for lung ADC. ACSL1, ACSL4, and ACSL5 are involved in lung ADC differentiation. ACSL5 may inhibit tumor proliferation in low-grade lung tumors, while ACSL3 may enhance proliferation in high-grade lung tumors. In addition, the application of ACSL inhibitors or ACSL knockdown may sensitize lung cancer cells to Gemcitabine or EGFR-TKIs. 

## Figures and Tables

**Figure 1 cancers-16-01170-f001:**
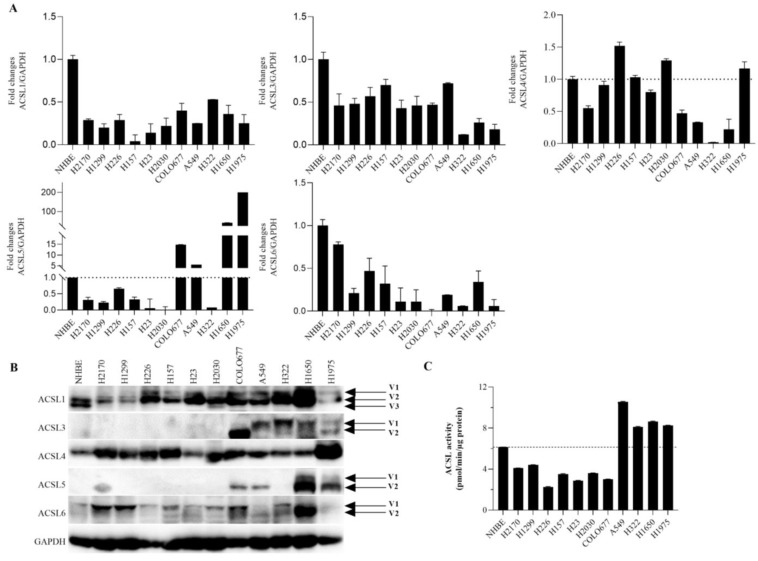
The expression of ACSL isoforms and ACSL enzymatic activity in normal bronchial epithelium (NHBE) cells and a panel of lung cancer cell lines. The expression of ACSL isoforms at both the mRNA and protein levels was analyzed by (**A**) RT-qPCR and (**B**) WB. Gene expression in comparison to the internal control GAPDH in NHBE cells was set to 1.0 for RT-qPCR analysis. GAPDH was used as a loading control for WB. (**C**) ACSL enzymatic activity was measured by liquid scintillation counting using [^3^H]-palmitic acid as the substrate. V: variant.

**Figure 2 cancers-16-01170-f002:**
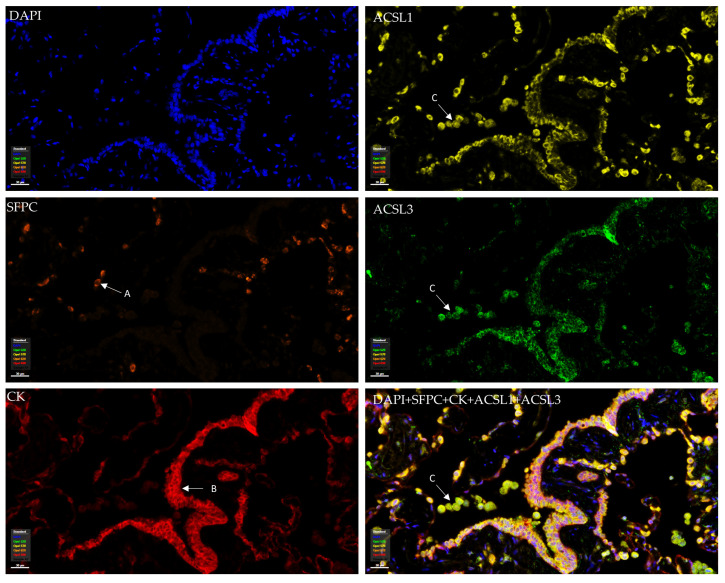
Tissue distribution and expression of ACSL1 and ACSL3 in normal human lung tissue. ACSL1 and ACSL3 were co-stained with cytokeratin (CK), surfactant protein C (SFPC), and DAPI in the same panel. (A) Alveolar type II cell. (B) Bronchial epithelial cells. (C) Macrophages.

**Figure 3 cancers-16-01170-f003:**
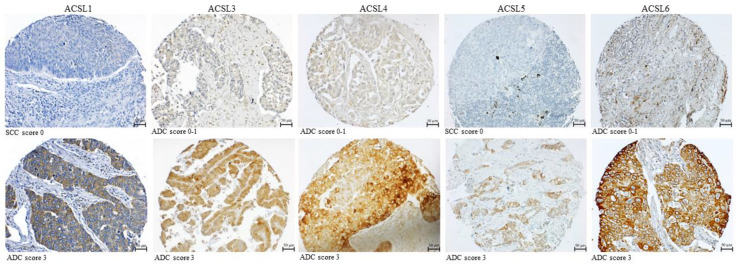
Representative staining of ACSL isoforms in primary lung tumors.

**Figure 4 cancers-16-01170-f004:**
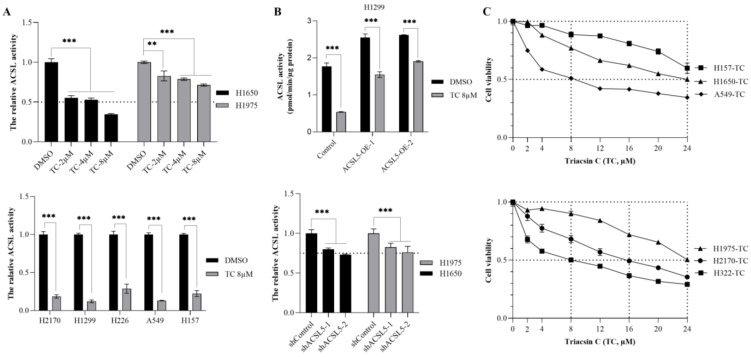
The effect of Triacsin C (TC) on ACSL enzymatic activity and cell viability in lung cancer cell lines. (**A**) The effect of TC (2, 4, and 8 µM) on ACSL activity in H1650 and H1975 (upper panel) cells; the effect of 8 µM of TC on ACSL activity in five lung cancer cell lines (lower panel). (**B**) The effect of TC (8 µM) on ACSL5-overexpressing cells (upper panel) and ACSL5-knockdown cells (lower panel); (**C**). The effect of TC at serial concentrations on cell viability in six lung cancer cell lines. ** *p* < 0.01, *** *p* < 0.001 when analyzed using Student’s *t*-test.

**Figure 5 cancers-16-01170-f005:**
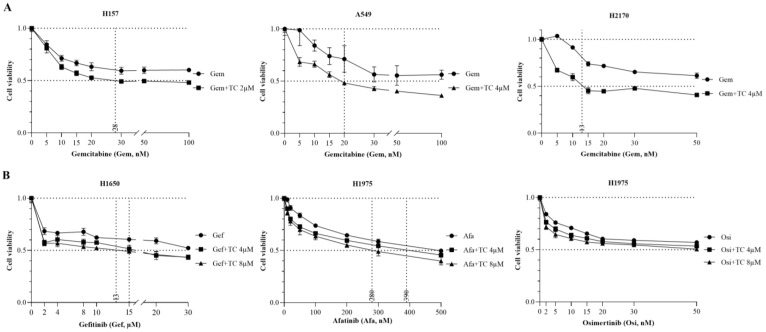
The antitumor efficacy of Triacsin C combined with Gemcitabine or EGFR-TKIs in lung cancer cell lines, as revealed by a cell viability assay. (**A**) The effect of Gemcitabine alone or combined with TC in both EGFR-WT cell lines. (**B**) The effect of EGFR-TKI alone or combined with TC in EGFR-mutant cell lines.

**Figure 6 cancers-16-01170-f006:**
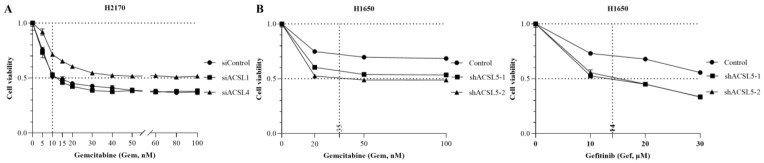
The effect of ACSL1, ACSL4, or ACSL5 knockdown combined with Gemcitabine or Gefitinib on cell viability in lung cancer cell lines. (**A**) The effect of ACSL1 and ACSL4 knockdown combined with Gemcitabine on cell viability in H2170. (**B**) The effect of ACSL5 knockdown combined with Gemcitabine (**left**) or Gefitinib (**right**) on cell viability in H1650.

**Table 1 cancers-16-01170-t001:** Association between the protein expression of ACSL family members and the clinicopathological parameters in primary lung tumors.

		Sample Number	ACSL1		*p*-Value	Sample Number	ACSL3		*p*-Value	Sample Number	ACSL4		*p*-Value	Sample Number	ACSL5		*p*-Value
			0	1–3			0–1	2–3			0–1	2–3			0	1–3	
Type	ADC	85	4	43	0.011	44	13	12	0.213	69	30	11	0.021	88	27	22	0
	SCC		12	26			14	5			27	1			38	1	
Gender	Male	97	18	57	0.387	53	29	12	0.09	78	48	12	0.501	101	62	16	0.069
	Female		3	19			5	7			16	2			14	9	
Age	<60	97	4	29	0.124	53	7	9	0.042	78	18	8	0.037	101	24	9	0.683
	≥60		17	47			27	10			46	6			52	16	
pT	1–2	93	18	64	1	49	24	16	0.454	75	56	10	0.345	97	61	23	0.504
	3–4		2	9			7	2			6	3			11	2	
pN	0–1	91	12	51	0.519	49	20	15	0.202	73	45	7	0.127	95	45	19	0.210
	2–4		7	21			11	3			15	6			26	5	
pM	0	97	18	69	0.685	53	30	17	1	78	58	12	0.629	101	70	22	0.686
	1–3		3	7			4	2			6	2			6	3	
Grade	1–2	97	6	43	0.023	53	14	13	0.057	78	39	3	0.009	101	32	17	0.025
	3–4		15	33			20	6			25	11			44	8	

ADC: adenocarcinoma; SCC: squamous carcinoma; T: tumor size; N: lymph node status; and M: metastasis.

**Table 2 cancers-16-01170-t002:** Correlation between protein expression of ACSL family members and tumor grade in primary lung SCC and ADC.

Type	Sample Number	Grade	ACSL1		*p*-Value	Sample Number	Grade	ACSL3		*p*-Value	Sample Number	Grade	ACSL4		*p*-Value	Sample Number	Grade	ACSL5		*p*-Value
			0	1–3				0–1	2–3				0–1	2–3				0	1–3	
ADC	47	1–2	0	25	0.026	25	1–2	4	9	0.027	41	1–2	20	3	0.024	49	1–2	10	15	0.03
		3–4	4	18			3–4	9	3			3–4	10	8			3–4	17	7	
SCC	38	1–2	6	16	0.503	19	1–2	10	3	1	28	1–2	17	0	0.393	39	1–2	21	1	1
		3–4	6	10			3–4	4	2			3–4	10	1			3–4	17	0	

ADC: adenocarcinoma; and SCC: squamous carcinoma.

**Table 3 cancers-16-01170-t003:** Correlation among the protein expression levels of ACSL family members in primary lung tumors.

50 Samples	ACSL1	ACSL3	ACSL4	ACSL5	ACSL6
ACSL1	1	r = 0.260, *p* = 0.069	r = 0.039, *p* = 0.788	r = 0.292, *p* = 0.04	r = −0.123, *p* = 0.394
ACSL3		1	r = −0.078, *p* = 0.589	r = 0.246, *p* = 0.085	r = −0.280, *p* = 0.048
ACSL4			1	r = −0.005, *p* = 0.972	r = −0.064, *p* = 0.659
ACSL5				1	r = −0.314, *p* = 0.026
ACSL6					1

r Value: correlation coefficient, *p*: Sig. (two-tailed).

**Table 4 cancers-16-01170-t004:** Correlation between the protein expression of ACSL family members and the proliferation marker Ki67 in primary lung tumors.

	ACSL1 vs. Ki67	ACSL3 vs. Ki67	ACSL4 vs. Ki67	ACSL5 vs. Ki67	ACSL6 vs. Ki67
Sample number	88	51	72	86	85
Correlation coefficient (r Value)	0.079	0.076	0.07	−0.15	−0.077
Sig. (two-tailed) *p*	0.458	0.585	0.551	0.164	0.477
Grade 1–2	44	26	38	42	44
	−0.122	−0.243	−0.293	−0.36	0.079
	0.42	0.234	0.071	0.02	0.598
Grade 3–4	44	25	34	44	41
	0.232	0.397	0.063	0.102	−0.241
	0.124	0.047	0.714	0.498	0.123

## Data Availability

Data are contained within the article and Appendix A.

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
