# Peer review of "The Diagnostic Value of ACSL1, ACSL4, and ACSL5 and the Clinical Potential of an ACSL Inhibitor in Non-Small-Cell Lung Cancer"

_cancers, 2024, doi:10.3390/cancers16061170_

Round 1
Reviewer 1 Report
Comments and Suggestions for Authors
The manuscript entitled "The diagnostic value of ACSL1, ACSL4 and ACSL5 and the clinical application of ACSL inhibitor in non-small cell lung cancer" by Y. Ma, M. Nenkov, A. Berndt, M. Abubrig, M. Schmidt, T. Sandhaus, O. Huber, J.H. Clement, S.M. Lang, Y. Chen, and N. Gassler describes the distribution of long chain acyl-CoA synthetase (ACSL) isoforms in various cancer cells and normal tissues as well as the role of this enzyme in metabolism of cancer cells. Various isoforms of ACSL are currently considered as a promising target for therapy and diagnostics of cancer. Despite the facts that information about the prevalence of ACSL isoforms in various tumor cells is a bit fragmentary, and the number of known ACSL inhibitors is very limited, research in this area opens the way to new modalities of diagnostics and adjuvant therapy.
The present study has been performed at very high scientific level, the results are clearly presented and are beyond any doubts. In vitro experiments has clearly demonstrated that combination of triacsin C in non-toxic dose with either Gemcitabine or EFGR-TKIs can be a plausible way to overcome drug resistance and to reduce side-effects of chemotherapy.
Given that the therapeutic potential of ACSL targeting for cancer treatment has been demonstrated only in in vitro experiments, I recommend to modify the title of the manuscript. In its present form, it refers to clinical application of triacsin C. At the same time, the data presented allow one to consider only the prospects (or perspectives) of its clinical application or enhancing the sensibility of lung cancer cells to chemotherapy. The title should not contain "the clinical application" itself.
Line 391: "knockdown" is typed as "knowdown".
After appropriate correction the title, the manuscript by Y. Ma and co-authors deserves publishing in the Cancers journal without further corrections.
Author Response
Dear reviewer,
First of all, we would like to thank you for taking time to review our manuscript and provide us with useful suggestions. Your inputs are highly appreciated! In the following part, please find our responses to the comments.
Given that the therapeutic potential of ACSL targeting for cancer treatment has been demonstrated only in in vitro experiments, I recommend to modify the title of the manuscript. In its present form, it refers to clinical application of triacsin C. At the same time, the data presented allow one to consider only the prospects (or perspectives) of its clinical application or enhancing the sensibility of lung cancer cells to chemotherapy. The title should not contain "the clinical application" itself.
- Yes, indeed, currently, the application of triacsin C in cancer treatment has been just at the preclinical stage (in cell lines and nude mouse xenograft models). The term “clinical application” for triacsin C is not very suitable. In the revised manuscript, we modified the title as „The diagnostic value of ACSL1, ACSL4 and ACSL5 and the clinical potential of ACSL inhibitor in non-small cell lung cancer”.
Line 391: "knockdown" is typed as "knowdown".
- Sorry for this typo. It has been corrected in the revised manuscript.
Reviewer 2 Report
Comments and Suggestions for Authors
In this manuscript, the authors showed the potency of the Acyl-CoA synthetase (ACSL) family as a diagnostic marker in non-small cell lung cancer. Furthermore, Inhibitors of ACSL may inhibit lung cancer proliferation.
It is interesting that ACSL is activated in cancer cells because they use fatty acids as energy.
Although this study showed that suppressing ACSL activity inhibits cancer cell proliferation, what kind of cell death is induced? Also, what happens in cancer cells due to changes in energy metabolism? The authors should present experiments to verify these points, or at least discuss them.
Author Response
Dear reviewer,
First of all, we would like to thank you for taking time to review our manuscript and provide us with useful advices. Your inputs are highly appreciated! In the following part, please find our responses to the comments.
Regarding the changes in energy metabolism, the following information has been included in the Discussion section (Line 509-513). Cancer cells utilizing the surround glucose, glutamin and lipids facillitate cancer cell proliferation and survival. Enhanced lipid metabolism accompanied by upregulated ACSLs has been seen in many cancer types. A growing evidence indicates that parmacological inhibition of ACSLs regulates lipid metabolism and incudes apoptosis and ferroptosis in different cancer types.
Regarding ACSL inhibition related the cell death, the information has been added into the Discussion section of the revised manuscript (Line 517-528). It has been depicted that TC could enhance etoposide-induced intrinsic apoptosis which was surppressed by overexpression of ACSL5 in glioma cells. Elevated ACSL4 accompanied by increased mitochondrial phospholipid biosynthesis and fatty acid oxidation promotes anti-apoptosis in chemoresistant breast cancer, and ACSL4 silencing surpresses tumor growth and induces apoptosis. Moreover, ACSL1, ACSL3 and ACSL4 are ferroptotic inducers. Inhibition of ACSL1-related ferroptosis by TC could restrain hyper-inflammation induced by coronavirus infection. Decreased ACSL3 activity by TC suppresses erastin-induced ferroptosis. Selective inhibition of ACSL4 by thiazolidinediones (TZDs), another PPARγ agonist, prevents RSL3 (ferroptocide)-induced ferroptosis and lipid peroxiation. Since ACSL4 promotes ferroptosis in lung cancer cells, it can be speculated that inhibition of ACSL4 by TC might result in reduced level of ferroptosis.
Best regards,
Reviewer 3 Report
Comments and Suggestions for Authors
Dear Authors!
I found your manuscript really interesting and well-written. You have used many lung cancer cell lines to demonstrate the diagnostic value of ACSLs. I think your manuscript is suitable for publication, but I would like to better explain in the discussion part (apart from alternative splicing) the paradox of decreased mRNA expression levels of ACSLs and the consequent enhanced translational expression levels.
Author Response
Dear reviewer,
First of all, we would like to thank all of the reviewers for taking time to review our manuscript and provide us with useful suggestions/advices. Your inputs are highly appreciated! In the following part, please find our responses to the comments.
The distinct mRNA and protein levels of ACSLs might be associated with somatic mutations, epigenetic mechanisms, and the post-transcriptional regulation which are frequently found in many biological systems. This information was mentioned in the Results section (Line 258), and now it is emphasized in the Discussion section (Line 424-426).
Best regards,